# Vaccine Coverage and Effectiveness in a School-Based Varicella Outbreak in Jinan Prefecture, Shandong Province

**DOI:** 10.3390/vaccines10081225

**Published:** 2022-07-31

**Authors:** Xiaoxue Liu, Quanxia Li, Xu Du, Xiaodong Zhao, Zundong Yin

**Affiliations:** 1Jinan Municipal Center for Disease Control and Prevention, No.2 Weiliu Road, Huaiyin District, Jinan 250021, China; freesia_xue@163.com; 2Licheng District Center for Disease Control and Prevention, Licheng District, Jinan 250199, China; jnslcqcdcjmk@jn.shandong.cn (Q.L.); lcmyyf-001@163.com (X.D.); 3Chinese Center for Disease Control and Prevention, No.27 Nanwei Road, Xicheng District, Beijing 100050, China

**Keywords:** varicella, school outbreak, vaccine, effectiveness, coverage

## Abstract

Background: Licheng District of Jinan Prefecture reported a school-based varicella outbreak. We conducted an investigation to analyze the epidemiology and scope of the outbreak, determine varicella vaccine coverage on the school campus, and estimate varicella vaccine effectiveness (VE). Methods: In the epidemiological investigation, we determined the attack rate, the clinical manifestations of varicella cases, and histories of prior varicella disease and varicella vaccination. We tested students for presence of serum IgM antibodies, and we attempted to isolate the varicella virus from vesicular fluid samples. We used chi-square to compare incidences between classes and floors. VE was estimated using a retrospective cohort study. Results: There were 13 varicella cases in the outbreak. All were among fourth grade students - twelve in Class 7 and one in Class 6. The attack rate in the two classrooms was 14.3% (13/91). Clinical symptoms were rash (100%) and fever (46.15%). All cases were reported within one average incubation period, and the epidemic curve suggested common exposure. Six of the 13 cases previously received one dose of varicella vaccine with a median time between vaccination and infection of 9 years; the other seven cases had not been vaccinated. Varicella vaccine coverage with one or more doses was 81.31%; 2-dose coverage was 38.15%. The median age of receipt of dose 1 was 1.18 years, and median age for receiving dose 2 was 5.12 years. One-dose varicella VE was 73.2% (95% confidence interval: 37.0%, 88.6%), and two-dose VE was 100%. Conclusions: Varicella vaccine coverage has been gradually increasing in recent years, as ≥1-dose and 2-dose coverage rates are higher in younger children than older children. High one-dose vaccination coverage limited the outbreak scope and led to the breakthrough cases being mild. Mild cases were difficult to detect in a timely manner. Varicella vaccine was highly effective, with 1-dose VE of 73% nine years after vaccination and 2-dose VE of 100%. We strongly recommended that all school students receive two doses of varicella vaccine.

## 1. Introduction

Varicella is a respiratory infectious disease caused by varicella-zoster virus (VZV) infection. It is transmitted through respiratory droplets or aerosols and direct contact with fresh vesicular fluid or mucosal secretions [1]. The incubation period is 10–21 days, most commonly 14–16 days [2]. Varicella occurs throughout the year. Although varicella is generally benign in children, serious complications can develop [3]. The main burden of varicella is economic due to the high number of cases and the need for parents and caregivers to take time to care for an ill child. Noncomplicated cases tend to last up to 2 weeks [4], during which time the child will not be able to attend day care or school.

In China, varicella is a nationally notifiable disease, and varicella school outbreaks are the leading cause of public health infectious disease emergencies. Varicella vaccine has been shown to be highly effective in preventing varicella disease [5], however it is not included in the National Immunization Program (NIP) and must be paid for by families in most cities. Based on theoretical and practical evidence [6,7,8], since 2011, Shandong Province has issued documents on Strengthening Varicella Prevention and Control, which suggest that eligible areas implement a two-dose varicella vaccine schedule with the first dose at 12 months and the second dose at 4–6 years. Because varicella vaccine is not included in Shandong Province’s local immunization planning program, its use in local areas is optional instead of compulsory, so high coverage is difficult to guarantee.

On 15 June 2021, Licheng District in Jinan Prefecture reported a varicella school outbreak that met the criteria of a public health emergency. The China Field Epidemiology Training Program (CFETP) and Licheng CDC conducted a joint investigation to manage the outbreak and prevent the spread of the varicella virus, assess varicella vaccine coverage, and estimate varicella vaccine effectiveness (VE). We report the results of the investigation.

## 2. Materials and Methods

### 2.1. Data Sources

We used the standardized varicella case questionnaire to record the clinical manifestations of the cases. Outbreak cases were identified in the China Disease Prevention and Control Information System and by asking teachers and school doctors about school absences. With teacher assistance, we used a custom-made questionnaire to record students’ varicella vaccination and disease histories. We used varicella vaccination documentation from the vaccination information management system to determine vaccination histories for students who could not provide official vaccination records. We obtained serum samples for IgM detection by the municipal laboratory. Vesicle fluid samples were sent to the provincial laboratory for virus isolation and culture. Data is contained within the article.

### 2.2. Case Definition and Classification

Case definitions were from the “Varicella Disease Surveillance Guide of Shandong Province.” Possible cases were students at the school between 16 May and 15 June 2021 with symptoms such as fever or characteristic varicella rash, or who were suspected by a medical doctor or hospital to have varicella. Confirmed cases were possible cases with one or more of the following: testing positive for varicella zoster IgM serum antibody and not vaccinated against varicella within one month; having convalescent serum varicella IgG antibody ≥4 times higher than in the acute stage; or having VZV isolated or VZV nucleic acid detected by polymerase chain reaction (PCR). According to the Standard on Information Reporting and Management of National Public Health Emergencies (Trial) [9], a varicella outbreak was defined as ≥10 cases of varicella occurring in a kindergarten or school within one week.

### 2.3. Case Search

We searched for cases by asking teachers and students about absences during the outbreak period, checking school attendance records with assistance from the teachers, and searching the China Disease Control and Prevention Information System for relevant varicella case reports.

### 2.4. Statistical Analysis

IBM SPSS Statistics 21 and Microsoft Excel 2010 were used for data processing and statistical analyses. We used standard descriptive statistics to describe the outbreak in terms of person, place, and time. Classroom and building floor attack rates were compared using the chi-square. Vaccine coverage trends were compared by grade with the chi-square trend test. Vaccine effectiveness (VE) was estimated by using a retrospective cohort study design. Risk ratios (RR) were compared using the chi-square test or Fisher’s exact probability method, and median times since vaccination were compared using rank-sum tests. VE was calculated as
(1)$$\mathrm{VE}=\frac{\mathrm{attack}\;\mathrm{rate}\;\mathrm{in}\;\mathrm{control}\;\mathrm{group}-\mathrm{attack}\;\mathrm{rate}\;\mathrm{in}\;\mathrm{vaccinated}\;\mathrm{group}}{\mathrm{attack}\;\mathrm{rate}\;\mathrm{in}\;\mathrm{control}\;\mathrm{group}}*100\%=\left(1-\mathrm{RR}\right)*100\%.$$

### 2.5. Ethical Review

Varicella is a nationally notifiable disease, and school outbreaks, including varicella school outbreaks, are nationally notifiable infectious disease emergencies. It is a public health responsibility to analyze routine disease surveillance data and to investigate outbreaks of infectious diseases. This study was a routine investigation and therefore exempt from ethical review.

## 3. Results

### 3.1. School Information

The affected school has two campuses (western campus and eastern campus) in Wanxiang community that are 1.5 km apart. The outbreak occurred in the western campus, which has 58 classes in 3 grade levels, 2557 students (44–46 per class), and 142 teachers. Most students were from surrounding communities and commuted to school. The classroom building has four floors; the first floor is for grade 2; the second floor is for grade 3, and the third and fourth floors are for grade 4. Each classroom has 6 windows and 2 doors, with good ventilation and lighting.

### 3.2. Epidemiology of Cases

There were 13 students in the outbreak setting who met the varicella case definition. The first case onset was on 7 June 2021 and the last was on 14 June 2021; all cases were reported or identified within a single VZV incubation period. Figure 1 shows the epidemiological curve, which is consistent with a point source outbreak. All clinical manifestations were mild, all 13 cases had a rash and 6 had fever; 9 were serum IgM positive. The outbreak was confined to grade 4 students: 5 boys and 8 girls, 10–11 years of age, all in fourth-floor classrooms. There were 12 cases in Class 7 and one in Class 6 (Figure 2). The attack rate in Class 7, the first class affected, was 14.3% (13/91); there was no pattern in the classroom seating of the cases.

### 3.3. Varicella Vaccine Coverage

Six of the 13 cases received one dose of varicella vaccine with a median time between vaccination and outbreak of 9 years; the seven others were not vaccinated. No cases reported previous varicella disease.

Table 1 shows vaccination coverage. Among all students on both campuses, ≥1 dose varicella vaccine coverage was 81.06%, and 2-dose coverage was 35.41%. Coverage was lower at higher grades (*p* = 0.00), as both one-or-more-dose and two-dose vaccination coverage rates were higher among young children compared with older children.

Coverage of the affected (western) campus with ≥1 dose was 81.31%, and 2-dose coverage was 38.15%. Vaccination coverage decreased with increasing grade (*p* = 0.00). In the affected campus, the median age of receipt of varicella vaccine dose one was 1.18 years (25th percentile 1.05, 75th percentile 1.50), and median age of receipt of dose two was 5.12 years (P25 4.24, P75 6.52). In the affected grade (Grade 4), the median age of varicella vaccine dose one was 1.18 years dose was 4.65 years. Vaccination rates varied by floor (Table 2), and median time since the last varicella vaccine dose of Class 8 was less than that of Class 7 (*p* = 0.039).

### 3.4. Vaccine Effectiveness

We estimated varicella VE among the students in Class 7, the most affected class. Class 6, with one case, was not included in the VE estimation. Table 3 shows the attack rate and VE by number of doses of varicella vaccine. There were 6 children who acquired varicella and had received one dose, with a median interval of 9 years between vaccination and the outbreak. There were 7 children who acquired varicella and had never been vaccinated against varicella. The attack rate among unvaccinated children was 77.8%; the attack rate among one-dose children was 20.8%; and the attack rate among 2-dose children was 0%. One-dose VE was 73.2%, and 2-dose VE was 100%.

### 3.5. Public Health Response

Affected classes were suspended for two weeks to control and isolate the source of infection. All cases of varicella were mild, and children were encouraged to isolate at home until the skin rash crusted over and the child was non-infectious.

Classroom windows were opened to improve ventilation. All classrooms were thoroughly disinfected, as were table surfaces, grounds, and door handles. Other grades and classrooms, canteens, and public places were also cleaned and disinfected.

Health monitoring was strengthened among teachers and students, including morning and afternoon check-ups, tracking absences from school, seeking medical advice and reporting fever or rashes. The epidemic situation was reported to local health and education departments, which were provided with regular progress reports.

Health education for parents, teachers, and students was enhanced to improve awareness of respiratory diseases such as varicella and influenza and how to prevent infection. Parents, teachers and students were told the necessity of receiving two doses of varicella vaccine to achieve maximal protection.

Checking varicella vaccination status during school enrollment was emphasized to schools and local community health centers. Emergency vaccination was not used during this outbreak because of the relatively high coverage of varicella vaccine among the students. Students who had not received two doses of varicella vaccine were encouraged to complete the two-dose series as soon as possible.

## 4. Discussion

There were 13 cases identified and reported in this varicella school outbreak. The outbreak was rapidly controlled by the measures taken, including suspension of classes that had cases; isolation, treatment, and observation of cases; and ventilation and disinfection of the campus environment. No further cases were identified after the control measures were implemented, even in the context of heightened respiratory screenings due to COVID-19 prevention and control measures. It is worth mentioning that the impending Dragon Boat Festival (a traditional festival with three days’ off work or school) contributed to the timely suspension and saved time for the subsequent investigation and management. The site investigation was on the second day of people returning from the holiday. Students of affected classes were required to stay at home for quarantine starting 12 June (the first day of the Dragon Boat Festival).

The scope of this outbreak was likely limited by high varicella vaccine coverage. Class 8 was adjacent to Class 7, which had all but one of the cases. No cases were found in Class 8, which may be related to high vaccination coverage and more recent vaccination in that class. The varicella vaccination coverage level we observed in this school was higher than most of cities in China [1], and only slightly lower than in Beijing and Tianjin where varicella vaccination is provided to children at no cost to families. Although Jinan does not provide free varicella vaccination to children, the high coverage found in this outbreak may be related to high parental awareness due to frequent campus outbreaks [2]. WHO recommends that if varicella vaccine is used, coverage should be maintained at 85% to 90% to most effectively protect individuals, achieve herd immunity, and avoid increasing the incidence of varicella in older individuals, who tend to have more severe cases. In our study, with 81.3% one-or-more-dose- and 38.2% two-dose-coverage, the outbreak was stopped within one incubation period and limited to only two adjacent classrooms. This result indicates that relatively high coverage can confine outbreaks and limit their scope. Additional studies, such as with the nonstandard finite difference method, are needed to help ascertain relationships between disease spread and other potential influencing factors such as heterogeneity, asymptomatic infection, and vaccination [10,11].

One dose of varicella vaccine is effective at protecting against moderate and severe varicella; however, a single dose has limited ability to prevent transmission and outbreaks [6]. Some studies show that outbreaks still occur even with ≥90% one-dose coverage [12,13,14,15]. These investigations highlight the need for a 2-dose varicella vaccination schedule. Not only do two doses provide better protection than a single dose, a two-dose schedule reduces varicella incidence markedly compared with a one-dose schedule [7,8]. In this outbreak, nearly half of the cases had only one dose of varicella vaccine, and no child who received two doses acquired varicella. We strongly recommend that the government introduce two doses of varicella vaccine into the immunization program as soon as possible. Vaccines that are included in the National Immunization Program rapidly achieve high coverage among their target populations. For varicella vaccine, such high coverage with two doses will help prevent school closures due to varicella outbreaks.

We found that the effectiveness of one dose of varicella vaccine was 73.2% nine years after vaccination. VE monitoring or estimation methodology includes randomized controlled trials, test-negative case-control studies, conventional case control studies, cohort studies, and ecological screening-method studies. Choice of study design depends on field conditions such as availability of data, identification of close contacts, availability of local vaccine coverage data, and availability of vaccination status data [16,17]. Disease transmission models can also be used to estimate vaccine efficacy during outbreaks or when surveillance identifies pathogen circulation [18]. There are many estimates of varicella VE made during outbreaks that use case-control or cohort study designs [19,20,21,22]. Retrospective studies are used widely in outbreak settings to rapidly estimate VE [23,24,25]. We used a retrospective cohort study design for our VE estimate due to high compliance with a powerful immunization information system that was able to provide accurate vaccination status data.

Among observational varicella VE studies, one-dose varicella VE ranges from 55% to 87%; two-dose varicella VE ranges between 84% and 98% [26,27,28]. VE is higher against moderate or severe disease than against infection [29,30,31]. In addition to influencing factors such as the number of administered doses, disease severity, and age at which the vaccine is administered [25,32], differences in VE may also be due to differences in baseline vaccination coverage [33], facility environmental hygiene, and prevention control measure implementation status [34]. In the context of a high vaccination rate, mild cases with subtle presentations may be difficult to detect, which could affect VE estimation [21]. During our investigation, some parents had misconceptions that their child had acne rather than varicella. This is consistent with mild illness and atypical symptoms associated with breakthrough cases. There were no complications among unvaccinated cases in this outbreak. Varicella is in general a mild disease, and there were too few cases to measure differences in severity by vaccination status. With high coverage and mild breakthrough cases, the amount of virus may be relatively low, resulting in fewer infections among those susceptible. Further research is warranted to understand the relationship between virulence and force of infection. Due to the small number of cases, it was not possible to conduct subgroup analysis of VE by vaccination year. It is therefore necessary to conduct studies on varicella VE based on different epidemic scopes, prevention and control levels, and vaccination coverage levels.

Our investigation was greatly facilitated by help from the teachers and parents and a powerful immunization and vaccination information system that provided vaccine histories. This allowed us to use a cohort study design with its greater power to evaluate VE than a case control study design. In addition to assessing coverage in classes with cases, we also obtained vaccine histories of other students in the school to assess herd immunity on this campus and determine risk of widespread epidemics and predict outbreak epidemiology. We reported coverage levels to the school administration to reinforce the necessity of checking vaccination records upon school enrollment and catching children in need of one or more doses of any vaccine. Readily obtained information including vaccine histories and in an affected school is an effective way to know herd immunization status and the risk of outbreaks. Such information can help schools and public health agencies determine appropriate actions in the face of measured risk. Vaccine histories were collected from parents, who were asked to check their children’s immunization records with the help and organization of class teachers. To avoid recall bias, parental vaccine histories that lacked exact dates were subsequently obtained through the immunization information system. In the future, with improved school enrollment inspection of vaccine record, the combination of immunization information system records with school attendance information system records can make school absence evaluations more precise and sensitive. Evaluating routine absence records is a way to identify childhood infectious diseases. We recommended to strengthen school absence registration system management to help detect cases in a timely manner through sensitive surveillance and improve health education and disease prevention and control awareness.

Our study was of a small outbreak in one school, which limited our ability to explore vaccine coverage across the prefecture and estimate VE by subgroups or over longer periods of time.

## 5. Conclusions

Varicella vaccine coverage has been gradually increasing in Jinan, as ≥1-dose and 2-dose of varicella vaccine coverage was higher in younger children than older children. High one-dose vaccination coverage led to breakthrough cases being mild and difficult to detect in a timely manner. Vaccine effectiveness of one dose was 73.2% nine years after vaccination; two-dose VE was 100%, as there were no cases among two-dose recipients. Varicella vaccine was highly effective at preventing varicella, with two doses being more effective than one dose. For preventing childhood morbidity and school closures due to varicella outbreaks, we strongly recommended that all school children receive two doses of varicella vaccine.

## Figures and Tables

**Figure 1 vaccines-10-01225-f001:**
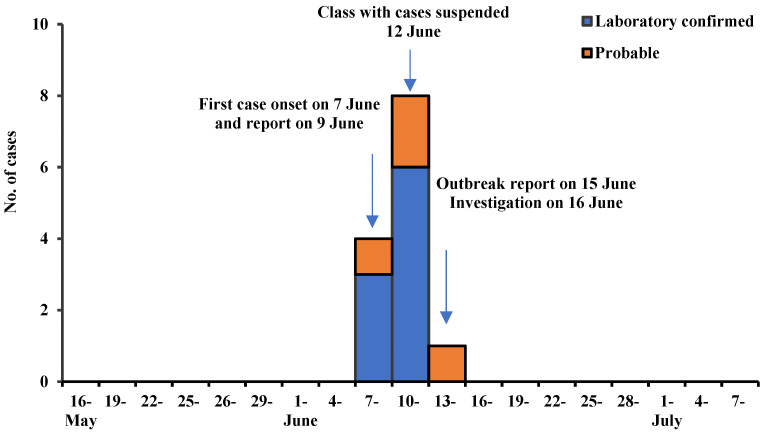
Time distribution of reported varicella cases.

**Figure 2 vaccines-10-01225-f002:**
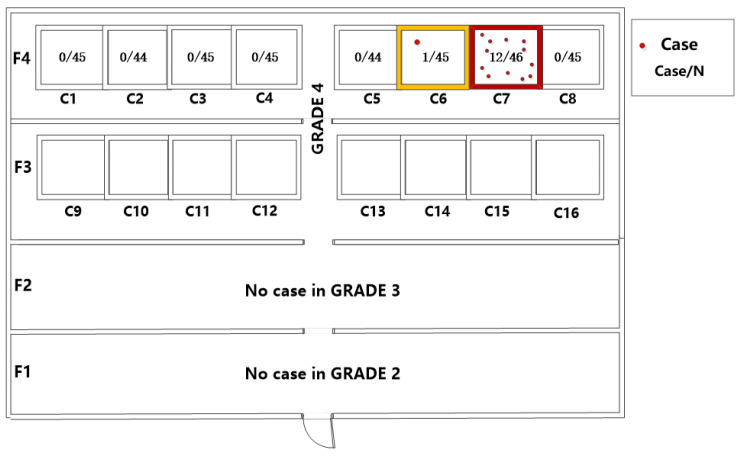
Classroom distribution of reported cases.

**Table 1 vaccines-10-01225-t001:** Varicella vaccination coverage on the two campuses.

	N	Varicella Vaccination History										
		≥1 Dose	1 Dose	2 Doses	None	Unknown	≥1 Dose (%)	Chi-square Trend	*p*	2 Doses (%)	Chi-square Trend	*p*
Eastern campus												
Grade 1	1077	936	321	615	113	28	86.91	56.65	0.00	57.10	511.00	0.00
Grade 5	675	547	377	170	97	31	81.04			25.19		
Grade 6	445	343	254	89	47	55	77.08			20.00		
Grade 7	408	297	239	58	91	20	72.79			14.22		
Grade 8	339	257	215	42	70	12	75.81			12.39		
Subtotal	2944	2380	1406	974	418	146	80.84			33.08		
Western campus (the affected school)												
Grade 2	927	791	348	443	114	22	85.33			47.79		
Grade 3	846	678	370	308	113	55	80.14			36.41		
Grade 4	720	558	358	200	131	31	77.50			27.78		
Subtotal	2493	2027	1076	951	358	108	81.31			38.15		
Total	5437	4407	2482	1925	776	254	81.06			35.41		

**Table 2 vaccines-10-01225-t002:** Varicella vaccine coverage by class on the affected floor.

	N	Varicella Vaccine History								History of Varicella Disease but Not Vaccinated
		≥1 Dose	1 Dose	2 Doses	None	Unknown	≥1 Dose (%)	2 Doses (%)	Time Since Last Dose (Years)	
Class 1	45	33	20	13	11	1	73.33	28.89	8.98	3
Class 2	44	34	23	11	7	3	77.27	25.00	8.76	2
Class 3	45	35	22	13	7	3	77.78	28.89	8.58	1
Class 4	45	36	22	14	9	0	80.00	31.11	8.41	0
Class 5	44	31	23	8	10	3	70.45	18.18	8.82	2
Class 6	45	35	25	10	10	0	77.78	22.22	8.83	1
Class 7	46	34	24	10	12	0	73.91	21.74	9.12	3
Class 8	45	37	19	18	5	3	82.22	40.00	7.25	1
Total	359	275	178	97	71	13	76.60	27.02	8.77	13

**Table 3 vaccines-10-01225-t003:** Varicella attack rate and vaccine effectiveness by number of doses of varicella vaccine.

Doses	N	Case	Attack Rate (%)	RR (95%CI)	VE (%, 95%CI)
0	9	7	77.8		
1	24	5	20.8	0.268 (0.114, 0.630)	73.2% (37.0, 88.6)
2	10	0	0.0	0	100%
Total	43	12	27.9		

## Data Availability

Data is contained within the article or Appendix A.

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
