# Peer review of "Vaccine Coverage and Effectiveness in a School-Based Varicella Outbreak in Jinan Prefecture, Shandong Province"

_vaccines, 2022, doi:10.3390/vaccines10081225_

Round 1

Reviewer 1 Report

In line 79, the term RT-PCR should be changed to PCR, since the varicella zoster virus genome is a double-stranded DNA and the retrotranscription process of its genetic material is not required before PCR.

In the Table 4. Varicella Attack Rates and Vaccine Effectiveness by Number of Doses of Varicella  Vaccine , How can a higher RR be explained in children with 1 dose of vaccine compared to >_ 1 as this measure is defined?, Likewise, when calculating the effectiveness of the vaccine, some mathematical formula must be thought of to correct the prevalence and effectiveness of the vaccine, which allows comparing the children who received >_ 1 dose with those who received 1 dose, I do not know if it would be that the interval between the single dose and the presence of chickenpox was shorter in this group compared to the children who received the dose, that is, the group of children with >_ 1 dose had a better immune response because their immune system was most recently stimulated?, When totaling the cases that presented Varicella , only 12 cases are mentioned, and throughout the document, 13 are mentioned, what happened to the missing case?

An explanation should be sought as to why none of the 7 cases that presented varicella  and were not vaccinated did not present any complications, and therefore this is not attributed to the effectiveness of the vaccine as the authors try to suggest.

Author Response

In line 79, the term RT-PCR should be changed to PCR, since the varicella zoster virus genome is a double-stranded DNA and the retrotranscription process of its genetic material is not required before PCR.

Reply: We appreciate the reviewer pointing out this mistake. We have made the correction throughout the manuscript.  

In the Table 4. Varicella Attack Rates and Vaccine Effectiveness by Number of Doses of Varicella  Vaccine , How can a higher RR be explained in children with 1 dose of vaccine compared to >_ 1 as this measure is defined?, Likewise, when calculating the effectiveness of the vaccine, some mathematical formula must be thought of to correct the prevalence and effectiveness of the vaccine, which allows comparing the children who received >_ 1 dose with those who received 1 dose, I do not know if it would be that the interval between the single dose and the presence of chickenpox was shorter in this group compared to the children who received the dose, that is, the group of children with >_ 1 dose had a better immune response because their immune system was most recently stimulated?, When totaling the cases that presented Varicella , only 12 cases are mentioned, and throughout the document, 13 are mentioned, what happened to the missing case?

Reply:Table 4 shows that a higher RR in children with 1 dose of vaccine compared to >_ 1. This makes intuitive sense as the risk of varicella is expected be higher in one-dose recipients than 2-dose recipients. Since VE is (1-RR)*100%, one would anticipate the lower RR to be associated with higher VE – and therefore higher VE among two-dose recipients.  To address the comment about the formulae, we made the statistical description more clear and formatted the VE equation.

For RR and VE calculations, we limited our analyses to the class that had all but one of the cases. We could not measure RR or VE reliably in a class with only one case. 

An explanation should be sought as to why none of the 7 cases that presented varicella and were not vaccinated did not present any complications, and therefore this is not attributed to the effectiveness of the vaccine as the authors try to suggest.

Reply: The reviewer brings up a very interesting point about our finding that even unvaccinated cases were mild. This is likely because varicella is generally a mild disease in childhood and the outbreak we studied had too few cases (13) to compare relative severity. We also think that there may be a relation to the attack rate among susceptibles (force of infection), which implies that there was less virus overall, likely due to high vaccination coverage and mild breakthrough cases.

In the Discussion section, we address this by stating, “In the context of a high vaccination rate, mild cases with subtle presentations may be difficult to detect, which could affect VE estimation [28]. During our investigation, some parents had misconceptions that their child had acne, rather than varicella. This is consistent with mild illness and atypical symptoms associated with breakthrough cases. There were no complications among unvaccinated cases in this outbreak. Varicella is in general a mild disease, and there were too few cases to measure differences in severity by vaccination status. With high coverage and mild breakthrough cases, the amount of virus may be relatively low, resulting in fewer infections among susceptibles. Further research is warranted to understand the relationship between virulence and force of infection. Due to the small number of cases, it was not possible to conduct subgroup analysis of VE by vaccination year. It is therefore necessary to conduct studies on varicella VE based on different epidemic scopes, prevention and control levels, and vaccination coverage levels.

Reviewer 2 Report

This study presents Varicella vaccine coverage and effective of vaccination during an outbreak in China. The major concerns from reading through this paper are as follows:

Understanding the basic determinants of vaccine uptakes would have enriched this study because there were just 12 cases of the diseases in a school of several thousands of students.

The presentation of the results should properly refer to the Tables and Figures.

Results in Tables 2 and 3 can be merged just as did in Table 1.

It is not clear how vaccine effectiveness was computed in Table 4. Can we say vaccine is effective without exposing the vaccinated to infection. This is pone of the basic themes of this study and unfortunately it is not clear at all.

Table 4 shows 12 cases instead of 13 that are reported in several parts of the paper.

In Table 4, cases for >=1 cases can be removed to avoid duplication.

Author Response

This study presents Varicella vaccine coverage and effective of vaccination during an outbreak in China. The major concerns from reading through this paper are as follows:

Understanding the basic determinants of vaccine uptakes would have enriched this study because there were just 12 cases of the diseases in a school of several thousands of students.

Reply: The reviewer makes a good point. We believe that high coverage helped limit the scope of the outbreak considerably. We now describe what we believe are reasons for high coverage, contrasting Jinan with Beijing and Tianjin strategies. This leads us to recommend a two-dose varicella vaccination schedule.

In the Discussion, we now say, “The scope of this outbreak was likely limited by high varicella vaccine coverage.  Class 8 was adjacent to Class 7, which had all but one of the cases. No cases were found in Class 8, which may be related to high vaccination coverage and more recent vaccination in that class. The varicella vaccination coverage level we observed in this school was higher than most of cities in China [1], and only slightly lower than in Beijing and Tianjin where varicella vaccination is provided to children at no cost to families. Although Jinan does not provide free varicella vaccination to children, the high coverage found in this outbreak may be related to high parental awareness due to frequent campus outbreaks [2].”

The presentation of the results should properly refer to the Tables and Figures.

Results in Tables 2 and 3 can be merged just as did in Table 1.

Reply: We corrected table/figure references. We prefer to keep table 3 (now table 2) separate because it provides the classroom view, which is different than the campus and grade view of coverage.

It is not clear how vaccine effectiveness was computed in Table 4. Can we say vaccine is effective without exposing the vaccinated to infection. This is pone of the basic themes of this study and unfortunately it is not clear at all.

Reply: To make VE calculation clear, we added the VE formula in method section. The statistical analysis section now states,

“IBM SPSS Statistics 21 and Microsoft Excel 2010 were used for data processing and statistical analyses. We used standard descriptive statistics to describe the outbreak in terms of person, place, and time. Classroom and building floor attack rates were compared using Chi-square. Vaccine coverage trends were compared by grade with the Chi-square trend test. Vaccine effectiveness (VE) was estimated by using a retrospective cohort study design. Risk ratios (RR) were compared using Chi-square test or Fisher's exact probability method, and median times since vaccination were compared using rank-sum tests. VE was calculated as VE=(attack rate in control group-attack rate in vaccinated group)/(attack rate in control group)*100%=(1-RR)*100%.

In order to avoid “vaccine is effective without exposing the vaccinated to infection”, we choose only the mostly affected class (class 7), and excluded class 6 with only one case to compute VE.

Table 4 shows 12 cases instead of 13 that are reported in several parts of the paper.

Reply: This is because we restricted VE analyses to the classroom with all but one case. We do not believe that a one-case classroom can be used for VE determination. In the VE section of the Results, we now state, “We estimates varicella VE among the students in Class 7, the most affected class. Class 6, with one case, was not included in the VE estimation.”

In Table 4, cases for >=1 cases can be removed to avoid duplication.

Reply: We have done so.

Reviewer 3 Report

I have some observations answer in needed before acceptance

1.The readability and presentation of the study should be further improved. 

2. The overall quality of the paper is in general, and the implementation aspect is clear.

3. The topic of this paper is of great interest. 

4. The superiority of the proposed approach must be compared with other established works.

5. The importance of the design carried out in this manuscript can be explained better than other important studies published in this field. I recommend the authors to review other recently developed works.

6. What makes the proposed method suitable for this unique task? What new development to the proposed method have the authors added (compared to the existing approaches)? These points should be clarified.

7.The performance of the proposed method should be better analyzed, commented and studied in more detail.

8. Furthermore, visit stochastic nonstandard finite difference method in the sense of modeling of diseases. This new research may be helpful to extend a future research (Find out in the literature). 

Author Response

I have some observations answer in needed before acceptance.

1. The readability and presentation of the study should be further improved. 

Reply: We have asked a native English speaker with subject matter knowledge for language editing to improve readability. He is acknowledged in the Acknowledgements section.

2. The overall quality of the paper is in general, and the implementation aspect is clear.

3. The topic of this paper is of great interest.

4. The superiority of the proposed approach must be compared with other established works.

Reply: We appreciate the comments on quality and interest. The reviewer’s point that the superiority of the approach must be compared with other approaches is a good point. We have included discussion about the methods in the Discussion section, stating, “There are numerous estimations of varicella VE during outbreaks that use case-control or cohort study designs [14,15,16]. Retrospective studies are used widely in outbreaks to estimate VE [17,18]. We used a retrospective cohort study design to estimate VE precisely. In general, one-dose varicella VE ranges from 55% to 87%; two-dose varicella VE ranges between 84% and 98% [19–21], and VE is higher against moderate or severe disease [22-24]. In addition to influencing factors such as the number of administered doses, disease severity, and age at which the vaccine is administered [19,25], differences in VE may also be due to differences in baseline vaccination coverage [26], facility environmental hygiene, and prevention control measures or implementation status [27]. In the context of a high vaccination rate, mild cases with subtle presentations may be difficult to detect, which could affect VE estimation [28].”

5. The importance of the design carried out in this manuscript can be explained better than other important studies published in this field. I recommend the authors to review other recently developed works. 

Reply: We agree that design considerations are important. We discuss briefly design in the reply to comment 4, stating, “There are numerous estimations of varicella VE during outbreaks that use case-control or cohort study designs [14,15,16]. Retrospective studies are used widely in outbreaks to estimate VE [17,18]. We used a retrospective cohort study design to estimate VE precisely. In general, one-dose varicella VE ranges from 55% to 87%; two-dose varicella VE ranges between 84% and 98% [19–21], and VE is higher against moderate or severe disease [22-24]. In addition to influencing factors such as the number of administered doses, disease severity, and age at which the vaccine is administered [19,25], differences in VE may also be due to differences in baseline vaccination coverage [26], facility environmental hygiene, and prevention control measures or implementation status [27]. In the context of a high vaccination rate, mild cases with subtle presentations may be difficult to detect, which could affect VE estimation [28].”

6. What makes the proposed method suitable for this unique task? What new development to the proposed method have the authors added (compared to the existing approaches)? These points should be clarified.

Reply: The purpose of our investigation was to provide evidence supporting varicella vaccination policymaking. Since VE estimates by number of doses is important, and since high coverage reduces force of infection, which can lead to not only fewer cases and milder cases, we added discussion points about herd immunity, mildness of unvaccinated cases in the context of high coverage and low force of infection, and factors that facilitated the VE investigation.

These points are made in the Discussion section as below.

“Our investigation was greatly facilitated by help from the teachers and parents and a powerful immunization and vaccination information system that provided vac-cine histories. This allowed us to use a cohort study design with its greater power to evaluate VE than a case control study design. In addition to assessing coverage in classes with cases, we also obtained vaccine histories of other students in the school to assess herd immunity on this campus and determine risk of widespread epidemics and predict outbreak epidemiology. We reported coverage levels to the school administration to reinforce the necessity of checking vaccination records upon school enrollment and catching up children in need of one or more doses of any vaccine. Readily obtained information including vaccine histories and in an affected school is an effective way to know herd immunization status and risk of outbreaks. Such information can help schools and public health agencies determine appropriate actions in the face of measured risk. Vaccine histories were collected from parents, who were asked to check their children’s immunization records with the help and organization of class teachers. To avoid recall bias, parental vaccine histories that lacked exact dates were subsequently obtained through the immunization information system. In the future, with improved school enrollment inspection of vaccine record, the combination of immunization in-formation system records with school attendance information system records can make school absence evaluations more precise and sensitive. Evaluating routine absence records is a way to identify childhood infectious diseases. We recommended to strengthen school absence registration system management to help detect cases in timely manner through sensitive surveillance and improve health education and dis-ease prevention and control awareness.”

And

“In addition to influencing factors such as the number of administered doses, disease severity, and age at which the vaccine is administered [19,25], differences in VE may also be due to differences in baseline vaccination coverage [26], facility environmental hygiene, and prevention control measures or implementation status [27]. In the context of a high vaccination rate, mild cases with subtle presentations may be difficult to detect, which could affect VE estimation [28]. During our investigation, some parents had misconceptions that their child had acne, rather than varicella. This is consistent with mild illness and atypical symptoms associated with breakthrough cases. There were no complications among unvaccinated cases in this outbreak. Varicella is in general a mild disease, and there were too few cases to measure differences in severity by vaccination status. With high coverage and mild breakthrough cases, the amount of virus may be relatively low, resulting in fewer infections among susceptibles. Further research is warranted to understand the relationship between virulence and force of infection. Due to the small number of cases, it was not possible to conduct subgroup analysis of VE by vaccination year. It is therefore necessary to conduct studies on varicella VE based on different epidemic scopes, prevention and control levels, and vaccination coverage levels.”

7.The performance of the proposed method should be better analyzed, commented and studied in more detail.

Reply: Some studies about VPD outbreaks evaluate VE according to different disease conditions (mild, severe) or different vaccine status (with only one dose, two doses or stratified analysis with the last dose received years since administration). While in our study, 13 cases (12 in class 7 and one in class 6) limited our ability to conduct stratified analysis.

We address this in the limitations paragraph in the Discussion section, stating, “Our study was of a small outbreak in one school, which limited our ability to explore vaccine coverage across the prefecture and estimate VE by subgroups or over longer periods of time.”

8. Furthermore, visit stochastic nonstandard finite difference method in the sense of modeling of diseases. This new research may be helpful to extend a future research (Find out in the literature). 

Reply: We appreciate the methodology advice. We find several literatures about stochastic nonstandard finite difference method in VPD or vaccination strategy, but still need time to learn about it and use it. If possible, could you please recommend one or two references about this method or its utilization (email: [email protected])?  We will learn from literature and try to use it in future VPD surveillance and bring this method to improve our knowledge in routine surveillance.

Round 2

Reviewer 2 Report

The privilege of reviewing this article is gratefully acknowledged. I found significant improvements in the revised version. However, there is the need to explain the ethical procedures that were followed under the methodology section. It is also not clear why the search started on January 1 and lasted till 15th June whereas the cases reported here were between 7-14 June. 

Table 1 needs to be reconciled because the figures there are not correct for some grades. The authors should recheck the values because figures under >=1 dose should be addition of 1 dose and 2 doses. These are not the case for some grades and the total number of pupil should be addition for 1 dose, 2 doses, none and unknown.

Author Response

The privilege of reviewing this article is gratefully acknowledged. I found significant improvements in the revised version. However, there is the need to explain the ethical procedures that were followed under the methodology section. It is also not clear why the search started on January 1 and lasted till 15th June whereas the cases reported here were between 7-14 June. 

Reply: We appreciate the positive feedback on the manuscript. The reviewer makes a good point about ethical review. Because this outbreak investigation was undertaken as a public health responsibility, it was exempt from ethical review. In response to the reviewer’s good point, we have added a new subsection called “Ethical review.” The text in that subsection says, “Varicella is a nationally notifiable disease, and school outbreaks, including varicella school outbreaks, are nationally notifiable infectious disease emergencies. It is a public health responsibility to analyze routine disease surveillance data and to investigate outbreaks of infectious diseases. This study was a routine investigation and therefore exempt from ethical review.”

The reviewer’s point about the timing of the data collection is also a good point. The search for outbreak-related cases started from May 16, which includes the longest likely incubation period before the first reported case. We used varicella cases reported in 2021 to analyze trends in the community.

To address the reviewer’s point about dates included in the outbreak investigation, we now state in the Methods section, “Possible cases were students at the school between May 16 and June 15, 2021 with symptoms such as fever or characteristic varicella rash, or who were suspected by a medical doctor or hospital to have varicella.”

Table 1 needs to be reconciled because the figures there are not correct for some grades. The authors should recheck the values because figures under >=1 dose should be addition of 1 dose and 2 doses. These are not the case for some grades and the total number of pupil should be addition for 1 dose, 2 doses, none and unknown.

Reply: We appreciate the reviewer pointing out this mistake. We rechecked the original history data and corrected the figures of Table 1 to ensure consistency and that the number of students with 1 or more doses is equal to the sum of the 1-dose and 2-dose students (there were no 3-or-more-dose students).

Reviewer 3 Report

The revised form is good.

Handle minor questions.

1.The performance of the proposed method should be better analyzed.

2. Literature should be enhanced. Visit literature.

3. Table of manuscript modify in the new form.

Author Response

1. The performance of the proposed method should be better analyzed.

Reply: We added information about methods for vaccine effectiveness estimations. We now justify our use of a retrospective cohort design for our VE analysis.  The new material in the Discussion section states, “VE monitoring or estimation methodology includes randomized controlled trials, test-negative case-control studies, conventional case control studies, cohort studies, and ecological screening-method studies. Choice of study design depends on field conditions such as availability of data, identification of close contacts, availability of local vaccine coverage data, and availability of vaccination status data [16,17]. Disease transmission models can also be used to estimate vaccine efficacy during outbreaks or when surveillance identifies pathogen circulation [18,19]. There are many estimates of varicella VE made during outbreaks that use case-control or cohort study designs [20-23]. Retrospective studies are used widely in outbreak settings to rapidly estimate VE [24-26]. We used a retrospective cohort study design for our VE estimate due to high compliance with a powerful immunization information system that was able to provide accurate vaccination status data.”

2. Literature should be enhanced. Visit literature.

Reply: We now include the review of VE methodology literature as described above. We also have added information on additional methodological work that is needed, stating in the Discussion section that “additional studies, such as with the nonstandard finite difference method, are needed to help ascertain relationships between disease spread and other potential influencing factors such as heterogeneity, asymptomatic infection, and vaccination [10,11].” Finally, we also review literature on other varicella VE studies in the Discussion section. We state, “Among observational varicella VE studies, one-dose varicella VE ranges from 55% to 87%; two-dose varicella VE ranges between 84% and 98% [27–29]. VE is higher against moderate or severe disease than against infection [30-32]. In addition to influencing factors such as the number of administered doses, disease severity, and age at which the vaccine is administered [26,33], differences in VE may also be due to differences in baseline vaccination coverage [34], facility environmental hygiene, and prevention control measure implementation status [35]. In the context of a high vaccination rate, mild cases with subtle presentations may be difficult to detect, which could affect VE estimation [36].”

3. Table of manuscript modify in the new form.

Reply: We reviewed and updated the tables. We believe they are in proper form now.